# Disentangled Acoustic Fields For Multi-modal Physical Property Understanding

## Abstract

We study the problem of multimodal physical property understanding, where an embodied agent needs to find fallen objects by inferring object properties, direction, and distance of an impact sound source. Previous works adopt feed-forward neural networks to directly regress the variables from sound, leading to poor generalization and domain adaptation issues. In this paper, we illustrate that learning a disentangled model of acoustic formation, referred to as disentangled acoustic field (DAF), to capture the sound generation and propagation process, enables the embodied agent to construct a spatial uncertainty map over where the objects may have fallen. We demonstrate that our analysis-by-synthesis framework can jointly infer sound properties by explicitly decomposing and factorizing the latent space of the disentangled model. We further show that the spatial uncertainty map can significantly improve the success rate for the localization of fallen objects by proposing multiple plausible exploration locations.

## 1 Introduction

Imagine walking through a forest with your eyes closed, listening to the sounds around you. As you move, you hear the rustling of leaves as an animal passes by, the gentle bubbling of a nearby stream, and the soft whisper of the wind. These sounds provide valuable information about the environment. Sound waves are influenced by the objects they encounter, changing in timbre, direction, and intensity as they reflect, diffract, and absorb. As humans, we intuitively understand how sound behaves in physical spaces, enabling us to infer the presence, location, and physical properties of objects from the sounds we hear.

Recent progress in neural fields has yielded high-fidelity models of perceptual modalities such as vision, touch, and sound. Most recently, neural acoustic fields (NAFs) (Luo et al., 2022) propose representing spatial acoustics of sounds, enabling continuous modeling of sound propagation and reverberation in a given scene. By modeling such acoustics, NAFs implicitly capture the structure and material properties of a scene. However, NAFs are overfitted to the acoustic properties of a single room, preventing them from being used as a disentangled model of sound across many environments.

In this work, we propose disentangled acoustic fields (DAFs), an approach to modeling acoustic properties across a multitude of different scenes. In NAFs, the short-time Fourier transform (STFT) of audio reverberation is the object of the modeling, but it is highly sensitive to the geometry of each scene and thus difficult to fit across different scenes. Instead, DAFs seek to model object sounds across multiple scenes using the power spectral density (PSD). This approach provides a lower dimensional compact representation of acoustics that preserves much of the physical information in emitted sounds. We demonstrate the effectiveness of this approach by showing high-accuracy object property inference across a set of different scenes.

We demonstrate how DAFs can be used to effectively enhance audio perception. Specifically, we propose using DAFs as a "mental simulation engine" that can test different physical property configurations to identify the world state that best matches the given sound. This "analysis-by-synthesis" approach allows us to robustly infer the underlying locations of fallen objects and effectively navigate to locate them. Our experiments show that, by using DAFs, we can accurately identify categories of fallen objects and their locations, even in complex acoustic environments.

Acoustic rendering with DAFs further enables us to obtain an uncertainty measure of different physical scene parameters, such as object locations, by assessing the mismatch between a simulated sound and ground-truth sound. We illustrate how such uncertainty may be used in the task of finding a fallen object, where we may naturally generate plans to different goals by considering the underlying uncertainty cost. In summary, our contributions are as follows:

- We introduce Disentangled Acoustic Fields (DAFs), an approach to model acoustic properties across a multitude of different scenes.
- We illustrate how analysis-by-synthesis using DAFs enables us to infer the physical properties of a scene.
- We illustrate how we may use DAFs to represent uncertainty and to navigate and find fallen objects.

## 2 RELATED WORK

### 2.1 NEURAL IMPLICIT REPRESENTATIONS

Learned implicit functions have emerged as a promising representation of the 3D geometry (Luo et al., 2021; Park et al., 2019; Xu et al., 2019), appearance (Mildenhall et al., 2021; Sitzmann et al., 2019), and acoustics of a scene (Luo et al., 2022). Unlike traditional discrete representations, implicit functions compactly encode information in the weights of a neural network, and can continuously map from spatial coordinates to output. Recent work has proposed to encode shapes as signed distance fields, learn appearance with differentiable rendering, and render acoustics by generating spectrograms (Xie et al., 2022) (Luo et al., 2022). For acoustics, Du et al. (2021) proposed to jointly generate acoustics and images by sampling from a joint manifold, and Luo et al. (2022) introduced the concept of Neural Acoustic Fields (NAFs), an implicit representation that captures sound propagation in a physical scene. While NAFs enable the modeling of sounds at novel locations in a single scene, they cannot be generalized to enable reasoning across novel scenes. In contrast, our method can generalize to novel scenes at test time, enables joint inference of object properties and location, and allows uncertainty-aware object localization.

### 2.2 MULTIMODAL SCENE UNDERSTANDING

Recent work has explored the use of input modalities beyond vision alone for scene understanding. Extensive studies have demonstrated the effectiveness of integrating audio and visual information in diverse scene understanding applications (Zhang et al., 2018; Owens et al., 2016b; 2018; Zhu et al., 2021). For instance, Ephrat et al. (2018); Zhao et al. (2018); Gao et al. (2018) employ visual input to separate and localize sounds, Gao et al. (2020) leverages spatial cues contained in echoes for more accurate depth estimation, while Chen et al. (2021c); Luo et al. (2022); Arandjelovic & Zisserman (2017); Owens & Efros (2018) demonstrate the potential of sound in learning multimodal features and inferring scene structure. Cross-modal generation has gained increasing attention by researchers (Gan et al., 2020a; Owens et al., 2016a; Su et al., 2020; Chen et al., 2017). Furthermore, Afouras et al. (2020); Gan et al. (2019); Arandjelovic & Zisserman (2018) integrate both visual and auditory information to localize target objects more accurately. Motivated by these findings, we propose a disentangled acoustic field for physical scene understanding, where an embodied agent seeks to find fallen objects by inferring their physical properties, direction, and distance from an impact sound.

### 2.3 AUDIO-VISUAL NAVIGATION

Our work is also closely related to audio-visual navigation, where navigation is achieved using audio signals to augment vision (Chen et al., 2021a; 2019; Gan et al., 2020c; Chang et al., 2017). In particular, Chen et al. (2020a) proposed the AudioGoal challenge, where an embodied agent is required to navigate to a target emitting a constant sound using audio for positional cues (Chen et al., 2020b). Building on this, Chen et al. (2021a) introduced the novel task of semantic audio-visual navigation, in which the agent must navigate to an object with semantic vision and short bursts of sound. However, their dataset had one limitation: it did not include synthesis capability for impact sound and thus could not render the physical properties (like material, position) by audio. To address both issues, Gan et al. (2022) proposed the Find Fallen Object task, where physical reasoning

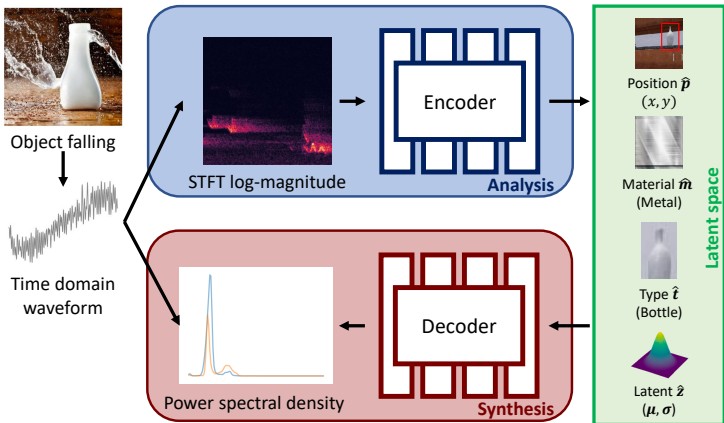

Figure 1: **Illustration of DAFs.** The encoder maps the binaural short-time Fourier transform (STFT) of the audio input into a new space containing physical audio information such as object position, material, type, and a continuous latent. The decoder utilizes these parameters to reconstruct the power spectral density (PSD) of the audio. The two components form an analysis-by-synthesis loop capable of inferring object properties, and are jointly learned during training.

was combined with sound. This dataset was based on the TDW (Gan et al., 2020b) simulation platform, which can generate audio from a variety of materials and parameters, and utilizes Resonance Audio (Google, 2018) (a technology for accurately replicating how sound interacts with the environment in 3D spaces) to spatialize the impact sounds depending on the room's spatial dimensions and wall/floor materials. Considering these advantages, we choose it as the benchmark to assess the capability of our proposed method on multi-modal physical property understanding.

## 3 PROPOSED METHOD

We are interested in learning a disentangled framework of sound that can effectively generalize across scenes, object types, and object locations. Key to our approach is an explicitly decomposed latent space that models the individual contribution of the sound factors. We first describe the parameterization of our disentangled acoustic field, which simultaneously enables factorization of the sound generation process and is defined on continuous locations in space. We further characterize the design choices that enable robust generalization and describe how we can use the continuous nature of our disentangled acoustic field to facilitate the localization of a fallen object.

### 3.1 PHYSICS OF SOUND

Given the sound of a falling object received by an agent as binaural signal $s$, we seek to identify the relative egocentric object location $p \in \mathbb{R}^3$, the object material category $m \in \{1, 2, \ldots, M\}$, the object type category $t \in \{1, 2, \ldots, T\}$, and a low-dimensional latent code $z \in \mathcal{R}^k$, where $z$ can contain information that is independent from the previous factors, such as information about the scene structure, scene materials, and properties about the propagation medium. Given an accurate model of sound formation $G$, we seek to reconstruct the sound via $G(p, m, t, z)$. In practice, we here do not reconstruct the sound itself, but its power spectral density, a simplified representation encompassing essential properties about the falling object.

### 3.2 DISENTANGLED ACOUSTIC FIELDS (DAFS)

We aim to learn a disentangled model of sound formation that facilitates efficient inference for characterizing sound properties. The parameterization of sound formation introduced in Section 3.1 provides a general framework for building such a disentangled model. To enhance effective learning, we structure our framework using an encoder, denoted as $E_\omega$, and a generator, denoted as $G_\phi$. By

instructing the network to consider the relative egocentric location of the sound emitter, we guide it to disregard the absolute positions of the agent and object, and reason in a generalizable fashion.

Given a sound signal represented as a binaural waveform $s \in \mathbb{R}^{2 \times t}$, we process the signal using the short-time Fourier transform (STFT), and retain the log-magnitude as $S$. Following prior work, we discard the phase present in the original signal, which is difficult to model (Du et al., 2021; Luo et al., 2022). We further investigated the choice of output representation, and found that the STFT of a fallen object sound used in prior work (Gan et al., 2022) includes large irregular stretches of uninformative silence along the time domain. The irregular and unpredictable temporal gaps are difficult for a neural network to effectively estimate and reconstruct, and in practice, a full-fidelity reconstruction of the sound signal in the original domain may not be necessary, as our ultimate goal is the inference of the underlying factors of sound formation. We thus transform the sound from the waveform domain into power spectral density (PSD) representation $\bar{S}$ using Welch's method (basically an average pooling over time of the squared STFT magnitude) as a target for our generator, which retains crucial information on the power present in each frequency, but collapses the information along the time dimension.

We may thus model the latent factors as the outputs of an encoder $E_\omega$ which takes as input the sound representation $S$:

$$(\hat{p}, \hat{m}, \hat{t}, \hat{\mu}, \hat{\sigma}) = E_\omega(S); \quad \hat{z} \sim \mathcal{N}(\hat{\mu}, \hat{\sigma}^2 \cdot \mathbb{I}), \tag{1}$$

where we model $\hat{z}$ as a sample from a diagonal Gaussian with mean $\hat{\mu}$ and standard deviation $\hat{\sigma}$. This restricted parameterization prevents the encoder from compressing all sound information into $\hat{z}$. The generator $G_\phi$ is modeled as a neural field which takes as input the latent factors and attempts to generate the PSD:

$$\hat{\bar{S}} = G_\phi(\hat{p}, \hat{m}, \hat{t}, \hat{z}). \tag{2}$$

We train our network with supervision on the latent factors and the output. For the $i$-th training example, we have access to the ground truth location, object material, and object type as the tuple $(p_i, m_i, t_i)$.

The object type and material are supervised with cross-entropy loss:

$$\mathcal{L}_{\text{type}} = \text{CrossEntropy}(\mathrm{t}_i, \hat{\mathrm{t}}_i), \tag{3}$$

$$\mathcal{L}_{\text{material}} = \text{CrossEntropy}(\mathrm{m}_i, \hat{\mathrm{m}}_i), \tag{4}$$

where $\mathrm{t}_i$ and $\mathrm{m}_i$ are the ground-truth object type and material for the $i$-th training sample, and $\hat{\mathrm{t}}_i$ and $\hat{\mathrm{m}}_i$ their estimates. An MSE loss is applied to facilitate the learning of the position vector:

$$\mathcal{L}_{\text{position}} = \frac{1}{2} \sum_i \|\hat{p}_i - p_i\|_2^2. \tag{5}$$

During training, we sample from the posterior for a given sound $S_i$ modeled as a multivariate Gaussian with diagonal covariance:

$$q_\omega(z|S_i) := \mathcal{N}(z; \mu_i, \sigma_i^2 \cdot \mathbb{I}) \tag{6}$$

We apply the reparameterization trick (Kingma & Welling, 2013) to allow for backpropagation through the parameters $\mu_i, \sigma_i$, by setting $z_i = \mu_i + \sigma_i \odot \epsilon$, where $\epsilon \sim \mathcal{N}(0, \mathbb{I})$. The latent $z$ is regularized with:

$$D_{\text{KL}}(q_\omega(z|S_i)\|\mathcal{N}(0, \mathbb{I})) \tag{7}$$

The output of the generator is supervised with an MSE loss to facilitate the prediction of the PSD:

$$\mathcal{L}_{\text{PSD}} = \frac{1}{d} \sum_i \left\| \hat{\bar{S}}_i - \bar{S}_i \right\|^2, \tag{8}$$

where $d$ is the dimension of the output PSD feature. In summary, our overall objective is to minimize:

$$\mathcal{L}_{\text{total}} = \alpha \mathcal{L}_{\text{type}} + \beta \mathcal{L}_{\text{material}} + \gamma \mathcal{L}_{\text{position}} + \delta D_{\text{KL}} + \eta \mathcal{L}_{\text{PSD}}, \tag{9}$$

where $(\alpha, \beta, \gamma, \delta, \eta)$ are hyperparameters.

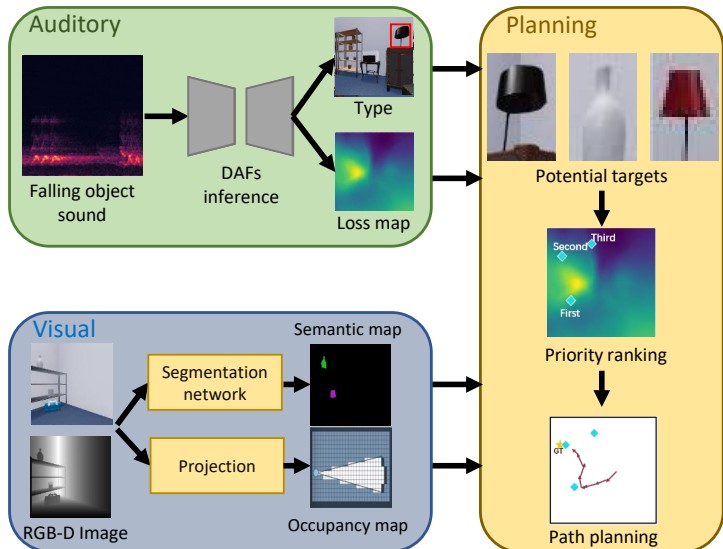

Figure 2: **Planning with DAFs.** The agent jointly uses auditory and visual information as part of the planning process. The auditory branch takes as input the sound $S$ represented as STFT. Using the DAF, we infer the factors responsible for the sound production including possible object types and a reconstruction loss map for each potential object location. The visual branch takes as input RGB-D images and provides a semantic map and occupancy map to the planner. The planner combines the information and uses the loss map to produce a priority list of locations. Path planning is completed using the $A^*$ algorithm.

### 3.3 INFERENCE OF SOUND PARAMETERS

We seek to invert the model of sound generation and compute the factors responsible for a given sound. Our disentangled model enables us to use "analysis by synthesis" to render all different acoustics parameters and find the one which matches our sound the best. However, the combinatorial complexity of enumerating combinations of factors renders this process computationally prohibitive. Instead, we amortize the inference of type, material, and latent into the joint learning process of the encoder and generator. We focus our efforts on the inference of object position, which is composed of continuous variables and is critical for the localization of the fallen object.

**Loss Map Generation:** Given a sound $s$ as recorded by an embodied agent, we use the encoder to infer the material, type, and continuous latent. We define a search space 10 m $\times$ 10 m centered on the agent position, and discretize this space using a resolution of 0.1 m. Using the previously inferred type, material, and Gaussian latent, we iterate over all possible locations $p_j$ where the object could be located. The current iterated position is combined with the other factors as inferred by the encoder network and provided to the generator.
The generated PSD is compared against the ground-truth PSD $\bar{S}$ of the sound of the fallen object, and an MSE difference is recorded at each location. In this fashion, we may generate a loss map corresponding to the two-dimensional coordinates. Since the loss map is based on the egocentric neural acoustic field, we need to convert the relative coordinates to the global frame of the room for path planning:

$$f_{r2g,c,\theta}\left(\left[\begin{array}{c} x \\ y \end{array}\right]\right) = \left[\begin{array}{c} x' \\ y' \end{array}\right] = \left[\begin{array}{cc} \cos\theta & -\sin\theta \\ \sin\theta & \cos\theta \end{array}\right]\left[\begin{array}{c} x \\ y \end{array}\right] + \left[\begin{array}{c} c_x \\ c_y \end{array}\right], \tag{10}$$

where $(c_x, c_y)$ is the agent's current position in the global frame and $\theta$ is its rotation angle in the global frame, while $(x, y)$ is the coordinate in the agent's egocentric frame.

**Uncertainty-Aware Planning:** We adopt a modular approach to path planning (Gan et al., 2022). Given the audio, we predict the object type and location via $E_\omega$. We further construct an internal

model of the world from RGB-D images of the environment. Semantic segmentation is computed using a pre-trained Mask-RCNN (He et al., 2017) model as illustrated in Figure 3. We use the depth map to project the semantic map into world space. Given the object types as inferred by the encoder network, we take the top-3 object type candidates as search targets. If a target is visible to the agent when initialized, the agent attempts to navigate to the location of the visible object. If there is no target in view, the agent will navigate to the predicted position. If the first attempt fails, the agent updates the world model and searches for potential object candidates. Object candidates are ranked according to the loss map value at the location corresponding to each object. Once the target list is determined, we apply the $A^*$ algorithm (Hart et al., 1968) to plan the shortest collision-free path to the first target in an unvisited area of the map.

---

**Algorithm 1:** Inferring a Loss Map of Positions

---

**Input:** Sound as log-magnitude STFT $S$ and PSD $\bar{S}$, encoder network $E_\omega$, generator network $G_\phi$, loss map grid $\mathcal{L}_{\text{grid}}$, function $f_{r2g,c,\theta}$ for global coordinates
1: $(\hat{p}, \hat{m}, \hat{t}, \hat{z}) = E_\omega(S)$
2: for $x_{\text{pos}}, y_{\text{pos}}$ in $[-5\,\text{m}, +5\,\text{m}]$:
3:      $\hat{p} = (x_{\text{pos}}, y_{\text{pos}})$
4:      $\mathcal{L}_{\text{grid}}[f_{r2g,c,\theta}(\hat{p})] = \|G_\phi([\hat{p}, \hat{m}, \hat{t}, \hat{z}]) - \bar{S}\|_2^2$

---

## 4 EXPERIMENT

### 4.1 INFERENCE OF OBJECT PROPERTIES

To test the physical inference capability of our proposed model, we first evaluate it on the Find Fallen Dataset (Gan et al., 2022) and compare it against two baselines. The first is the modular sound predictor presented in Gan et al. (2022), which was learned without the use of a disentangled model. The second is a gradient-based optimization process that minimizes the difference between the predicted and ground-truth PSD by optimizing all latent factors using our trained generator. All methods are evaluated on the same test split. To enable exploring multiple plausible locations, we mark a type as accurately predicted if the correct object type is within the top-3 predicted categories. The results in Table 1 show that our model significantly outperforms the baseline methods in both position and type prediction accuracy. By jointly learning a disentangled model along the encoder, we can more accurately predict the object location and the object type. Gradient-based optimization fails in jointly optimizing the position and object type, and is easily stuck in local minima.

To further evaluate our model in real-world scenarios, we employ the REALIMPACT dataset (Clarke et al., 2023), encompassing 150,000 impact sound recordings across 50 real-world object categories. This dataset incorporates impact sounds captured from 600 distinct listener locations, spanning 10 angles, 15 heights, and 4 distances. We use the same train/test split across all methods. To accommodate with the dataset, we adapt the output of the encoder network and the input of the generator network to be angle, height, and distance. The official baseline method is a ResNet-18 network employing the magnitude spectrogram of impact sounds to predict the corresponding object properties. As highlighted in Table 2, our method demonstrates a significant improvement over the baseline in predicting the category of object angle, height, and distance.

Table 1: **Comparison of position error and type prediction accuracy on Find Fallen.** Our predictor learned alongside the disentangled model is significantly more accurate in both position and object type (top-3) prediction. We observe that gradient-based optimization of these factors using the disentangled model does not succeed.

| Method | Position Error (m) ↓ | Type Acc. ↑ |
|---|---|---|
| Modular Predictor Gan et al. (2022) | 2.41 | 0.32 |
| Gradient Inversion | 3.19 | 0.11 |
| Our Predictor | **1.09** | **0.84** |

Table 2: **Comparison of angle, height, and distance category prediction accuracy on RE-ALIMPACT dataset.** Our method significantly outperforms the baseline method in all three tasks.

| Method | Angle Acc. ↑ | Height Acc. ↑ | Distance Acc. ↑ |
|---|---|---|---|
| Chance | 0.100 | 0.067 | 0.250 |
| Baseline Gao et al. (2022) | 0.758 | 0.881 | 0.983 |
| Ours | **0.900** | **0.960** | **0.994** |

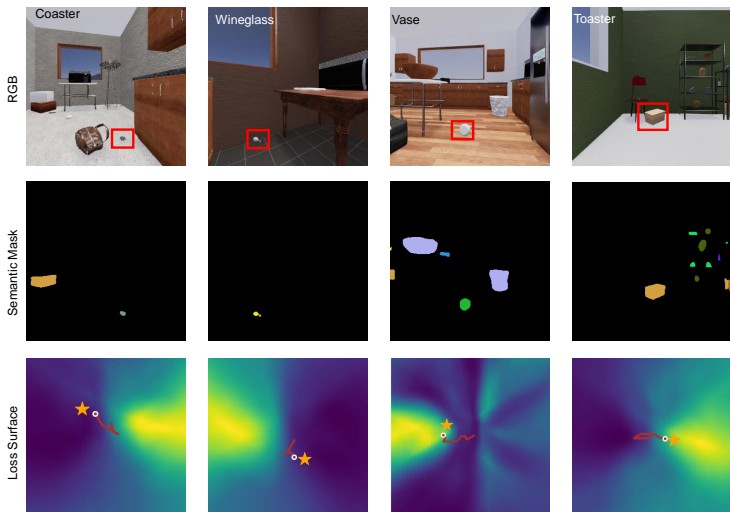

Figure 3: **Visualization of visual input and the sound-derived loss map in four scenes. Top:** RGB images of the agent's view with the target object in a red bounding box. **Middle:** Semantic map produced from the RGB images. **Bottom:** The red line indicates the path the agent takes, with the end point shown as a circular dot. The ground-truth object location is shown as a gold star.

## 4.2 NAVIGATION AND PLANNING

**Experimental Setup.** We use the TDW (Gan et al., 2020b) simulation platform to evaluate our proposed method. Our networks are trained on the Find Fallen Object dataset[1], following the experimental configuration described in Gan et al. (2022). This dataset contains 8000 instances of 30 physical object types in 64 physically different rooms (32 study rooms and 32 kitchens), providing a wide range of acoustic scenarios for evaluation. We evaluated the models' performance on the test split identified by Gan et al. (2022). The audio was only available at the beginning of the test, and the agent would receive an RGB-D image at every subsequent step. For the available actions, we set move forward to 0.25 m and rotate to 30 degrees. The task was defined as follows: an embodied agent with an egocentric-view camera and microphone hears an unknown object fall somewhere in the room it is in (a study room or kitchen) as shown in Figure 3; the agent is then required to integrate audio-visual information to find which object has fallen and where it is, as efficiently as possible. Audio is recorded at 44.1 kHz in a two-channel configuration. We generate the STFT representation using a window and FFT size of 512, a hop length of 128, and a Hann window. The PSD representation is generated using Welch's method with a Hann window of size 256, an overlap of 128, and FFT size of 256.

We evaluate agents using three metrics: Success Rate, Success weighted by Path Length (SPL) (Anderson et al., 2018), and Success weighted by Number of Actions (SNA) (Chen et al., 2020b). The Success Rate is calculated as the ratio of successful navigation trials to the total number of trials. A trial is considered successful if the agent explicitly executes action found when the distance between the agent and the object is less than 2 meters, the target physical object is visible in the agent's view, and the number of actions executed so far is less than the maximum number of allowed

---

[1]https://github.com/chuangg/find_fallen_objects

steps (set to 200 in all tests). SPL is a metric that jointly considers the success rate and the path length to reach the goal from the starting point. SNA takes into account the number of actions and the success rate, penalizing collisions, rotations, and height adjustments taken to find the targets.

Table 3: **Comparison against baseline methods on the localization of fallen objects.** We find that our full model which utilizes a disentangled model to produce an uncertainty surface is significantly improved when evaluated on SR, SPL, and SNA metrics. Ablation studies which progressively replace parts of the original method with ones trained along our disentangled model show the contribution of each component. Baseline results are taken from Gan et al. (2022).

| Method | SR ↑ | SPL ↑ | SNA↑ |
|---|---|---|---|
| Decision TransFormer (Chen et al., 2021b) | 0.17 | 0.12 | 0.14 |
| PPO (Oracle found) (Schulman et al., 2017) | 0.19 | 0.15 | 0.14 |
| SAVi (Chen et al., 2021a) | 0.23 | 0.16 | 0.10 |
| Object-Goal (Chaplot et al., 2020) | 0.22 | 0.18 | 0.17 |
| Modular Planning (Gan et al., 2022) | 0.41 | 0.27 | 0.25 |
| Modular Planning + Loss Map | 0.43 | 0.30 | 0.29 |
| Modular Planning + Our Position | 0.44 | 0.29 | 0.28 |
| Modular Planning + Our Type | 0.51 | 0.34 | 0.34 |
| Ours (Full model) | **0.57** | **0.38** | **0.37** |

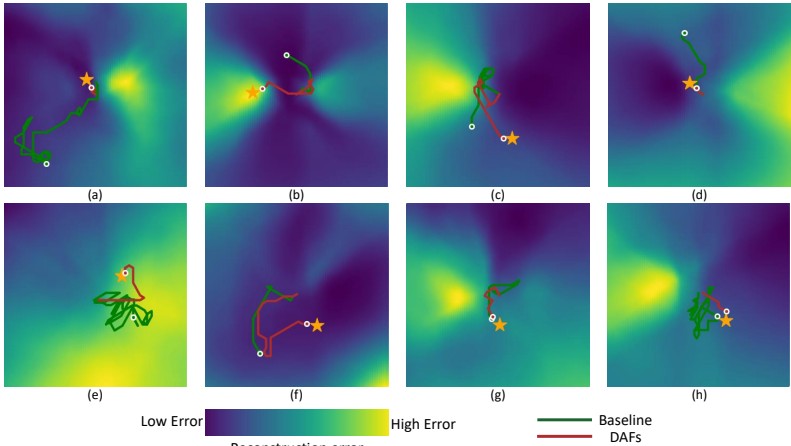

Figure 4: **Comparison of agent trajectories.** We compare the agent trajectories using our method (Red) against the trajectories produced by the modular planning baseline (Green). The loss map uses dark blue to indicate regions of low error, while yellow is used to indicate regions of high error. This figure compares the uncertainty maps of various cases. Darker colors indicate lower values of position loss. The star (Gold) symbolizes the ground truth position of the fallen object. The end of each trajectory is circled in white for clarity. In (a)−(f), the baseline method fails to find the target, while our method succeeds. In (g)−(h), both methods find the target, but our method takes a shorter path.

**Result analysis.** We evaluate the effectiveness of our proposed method against strong baselines as proposed in Gan et al. (2022). In Table 3, we find that our disentangled model based method can significantly outperform the previous modular planning baseline by 14% absolute in success rate. We further observe a significant decrease in the average length of the path and the number of actions taken by the agents. In addition to comparing our full proposed method with the previous modular planning baseline, we conduct an ablation study to assess the necessity of introducing decoders that generate audio signals from object properties. Specifically, we evaluate a variant of our method that navigates to the location predicted by the encoder. This alternative approach shows comparable success rates, but with lower SPL and SNA metrics, emphasizing the critical role of the loss map in effective exploration. We visualize the loss map and trajectories taken by the agent in Figure 4. We observe in Figure 4 (a)∼(f) that the modular planner often fails to find the target and attempts to find

the object via random search, while our method can plan a direct route to the object. In Figure 4 (g)∼(h), both methods find the target, but our method takes a shorter path. These results illustrate the superiority of our proposed approach over the baseline modular planner.

**Ablation studies.** The modular nature of our proposed method facilitates ablation studies that yield insight into the contribution of each individual component. We report results for ablations in Table 3. Beginning with the modular planning baseline, we find that augmenting the planner with a loss-map-guided object priority ranker yields a 2% increase in SR, a 3% increase in SPL, and 4% increase in SNA. This shows that the introduction of the uncertainty map can effectively improve the efficiency of agents searching for potential objects, reducing both the length of the path and the number of actions taken. Additionally, we replaced the sound location and sound type predictors in modular planning with our predictor jointly trained with a generator. The improvement in the object type prediction accuracy was found to contribute more to the overall SR than the improvement in the position accuracy. This result corroborates the conclusion in Gan et al. (2022) that accurately predicting object types from audio alone is a major challenge.

Table 4: **Evaluation of cross-scene prediction for DAFs.** Compared to Table 1, there is a small decrease in accuracy.

| Scene | Position Error (m)↓ | Type Acc.↑ |
|---|---|---|
| Kitchen to Study Room | 1.17 | 0.81 |
| Study Room to Kitchen | 1.23 | 0.80 |

Table 5: **Evaluation of cross-scene generalization of different methods.** We find that all methods will experience a decrease in success rate as evaluated by different metrics. However, by analyzing the sound information using a disentangled framework, our DAFs can robustly generalize to novel scene types unseen during training. Baselines are taken from Gan et al. (2022).

| Method | Kitchen to Study Room | | | Study Room to Kitchen | | |
|---|---|---|---|---|---|---|
| | SR ↑ | SPL ↑ | SNA ↑ | SR ↑ | SPL ↑ | SNA ↑ |
| PPO (Oracle found) | 0.11 | 0.10 | 0.10 | 0.05 | 0.04 | 0.05 |
| SAVi | 0.20 | 0.11 | 0.09 | 0.19 | 0.14 | 0.11 |
| Decision TransFormer | 0.07 | 0.06 | 0.06 | 0.08 | 0.06 | 0.07 |
| Object Navigation | 0.18 | 0.14 | 0.13 | 0.15 | 0.14 | 0.13 |
| Modular Planning | 0.34 | 0.23 | 0.20 | 0.35 | 0.22 | 0.19 |
| Ours | **0.52** | **0.38** | **0.37** | **0.48** | **0.32** | **0.32** |

**Cross-Scene Generalization.** To explicitly assess the generalization ability of our proposed method, we train and test on entirely different classes of rooms. In the first split, models are trained in the kitchens and tested in the study rooms. For the second split, models are trained in the study rooms and tested in the kitchens. The object property prediction results are reported in Table 4. In both splits, the accuracy of positioning and predicting the type of object slightly decreased compared to that of the full-trained model. The planning results are reported in Table 5, where all models experience a degree of cross-scene performance drop. The success rate of the modular planning approach decreases by 7% in SR on the first split, while our method only decreases by 4%. Our proposed method still performs the best in both splits. This highlights that our method can not only generalize across room instances of the same type, but can also effectively generalize across rooms of a different type.

## 5  CONCLUSION

This paper presents an egocentric disentangled acoustic field framework that can generalize and reason across scenes. Joint inference of sound properties is implemented by using an explicit decomposition in the latent space. Furthermore, our approach is capable of generating multimodal uncertainty maps. Experiments on the TDW simulation platform demonstrate our disentangled acoustic field can improve the success rate for the localization of fallen objects. Consequently, our proposed method is a promising solution for sound localization and understanding in complex scenes.

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

# A  APPENDIX

In Section A.1, we conduct a comparative experiment against NAF in the task of audio synthesis. In Section 7, we test two more strong baseline methods on the REALIMPACT dataset(Clarke et al., 2023) to evaluate the performance on sound source localization. In Section A.3, we evaluate the inference ability of our model on a novel dataset. In Section A.3, we analyze some failure cases in the navigation task. In Section A.5, we ablate the effect of using PSD compared to STFT for acoustic modeling, finding that the PSD representation is easier for the network to reconstruct and yields higher type and position prediction performance. In Section A.6, we perform an experiment using ground-truth semantic segmentation, which helps illustrate the upper bound performance of our method.

## A.1  COMPARATIVE EXPERIMENT AGAINST NAF

To conduct a comparative experiment against NAF, we implement a baseline model equivalent to NAF. We then conducted a comparative analysis of our approach and NAF using 15,000 audio files from the REALIMPACT dataset (Clarke et al., 2023). This analysis focused on the accuracy of audio simulation, measured by comparing the Mean Squared Error (MSE) of the reconstructed Power Spectral Density (PSD) against the true PSD. The results, detailed in Table 6, clearly demonstrate our method's superior performance in audio simulation tasks compared to NAF.

Table 6: Comparison of PSD reconstruction accuracy on REALIMPACT dataset (Clarke et al., 2023).

| Method | NAF | Ours |
|---|---|---|
| PSD MSE ↓ | 0.0264 | **0.0052** |

## A.2  SOUND SOURCE LOCALIZATION EVALUATION

we choose two strong methods on sound source localization methods (Zhu et al., 2021) and implement them with pytorch. One feeds the STFT of the audio into U-Net Patel et al. (2020), and the other feeds the original waveform into CNN (He et al., 2021). To evaluate the performance of our model in sound source localization, we employed the REALIMPACT dataset(Clarke et al., 2023). The experimental results, presented in the table , clearly demonstrate that our proposed model outperforms the established baseline methods. We appreciate the guidance to substantiate our claims through comparative analysis and believe these additional results strengthen the effectiveness of our proposed approach.

Table 7: Comparison of angle, height, and distance category prediction accuracy on REALIMPACT dataset against some cutting-edge methods.

| Method | Angle Acc. ↑ | Height Acc. ↑ | Distance Acc. ↑ |
|---|---|---|---|
| Chance | 0.100 | 0.067 | 0.250 |
| U-net+STFT (Patel et al., 2020) | 0.825 | 0.902 | 0.972 |
| CNN+waveform(He et al., 2021) | 0.671 | 0.755 | 0.802 |
| Resnet+STFT (Clarke et al., 2023) | 0.758 | 0.881 | 0.983 |
| Ours | **0.900** | **0.960** | **0.994** |

## A.3  EVALUATION ON OBJECTFOLDER2 DATASET

We evaluate our model on the OBJECTFOLDER2 Gao et al. (2022) dataset, which contains 1000 virtualized objects with different sizes and types along with acoustic, visual, and tactile sensory information. Correspondingly, we adjust the output of the encoder network and the input of the decoder network to correspond to the object scale and type. As indicated in Table 8, our method surpasses the performance of the baseline approach Gao et al. (2022), which employs a ResNet-18 architecture utilizing the magnitude spectrogram as input to predict the object scale and type.

Table 8: **Comparison of scale estimation and type prediction accuracy on OBJECTFOLDER2.**

| Method | Scale (m) ↓ | Type Acc. ↑ |
|---|---|---|
| Baseline Gao et al. (2022) | 0.20 | 0.98 |
| Ours | **0.17** | **0.99** |

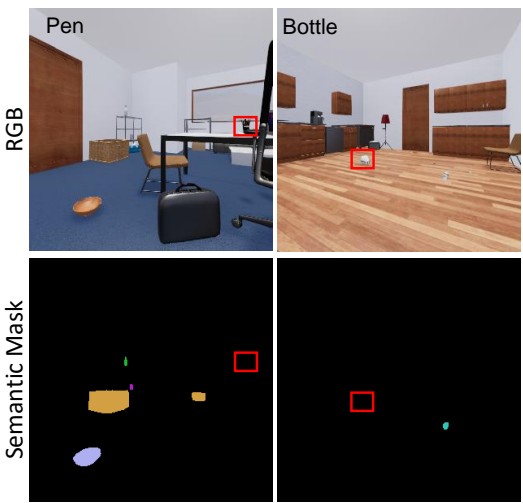

Figure 5: **Failure from visual branch.** The visual branch is learned independently of the auditory branch. Semantic segmentation errors can occur when objects are visually small or of low contrast. Future work can explore the contrastive learning of joint audio-visual representations.

### A.4 NAVIGATION FAILURE CASE ANALYSIS

Despite significant performance improvements upon baseline methods in the navigation task, our method still may sometimes fail to find the fallen object. Upon analyzing some failure cases, we discover that inaccurate semantic segmentation is one major problem. In such scenes, even if the target position and type are accurately predicted by our audio network, the agent would not be able to find the object. As illustrated in Figure 5, some segmentation failures are due to the object being too small, or its color being too close to the background color. Additionally, distractor objects may cause the agent to use the `found` command on the wrong object. We show two trajectories where our method cannot find the object or uses a long path. Figure 6 (a) shows a case where multiple distractors of the same kind as the fallen object are in the agent's view, and thus both the baseline and our method fail to navigate to the target location. As seen in Figure 6 (b), the introduction of a loss map does not necessarily ensure that the agent takes the shortest path. Our method succeeds in finding the target, but it takes a longer path than the baseline.

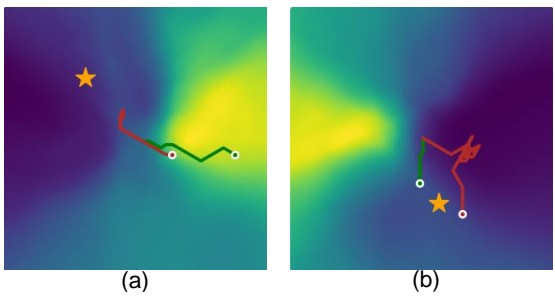

Figure 6: **Trajectory of failure cases.** **(a)** Both our method and the modular planning baseline execute the `found` command on the wrong object. **(b)** Our method takes a longer path than the baseline method and searches a low error region of the loss map.

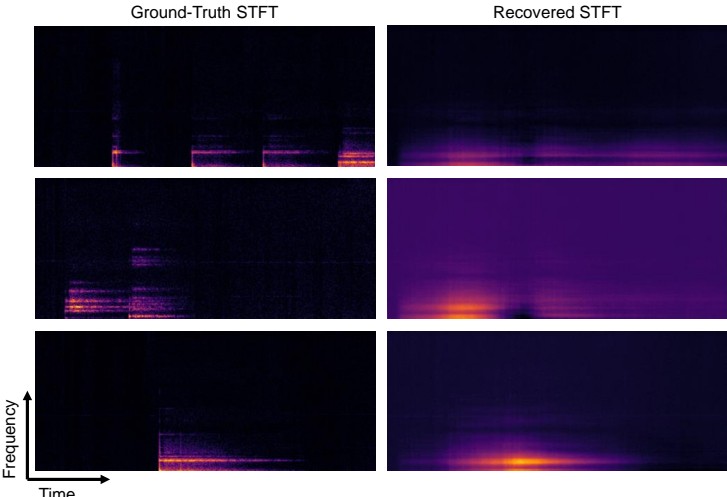

Figure 7: **Visualization of STFT Reconstruction. Left:** The ground-truth log-STFT of an impact sound. **Right:** The log-STFT recovered by a network modified to output STFT instead of PSD. In the three examples shown here, we find that the network struggles to model the irregular temporal caps present in the STFT.

## A.5    CHOICE OF OUTPUT REPRESENTATION

In the main paper, we choose to utilize power spectral density (PSD) as the choice of output representation, instead of the short-time Fourier transform (STFT). The PSD representation captures the power in each specific frequency, but unlike STFT, it discards the temporal information (PSD is similar to an average pooling across time of the squared STFT magnitude).

In this experiment, we seek to reconstruct the STFT. We utilize the network parameterization proposed by Neural Acoustic Fields (NAFs). The STFT was supervised with an MSE loss. After training, we evaluate the two models on a test set of 100 instances. The results of the object type and location predictions are shown in Table 9, where the predictor supervised by PSD reconstruction achieves higher accuracy in both object type and position. Figures 7 and 8 compare the ground-truth and recovered values of STFT and PSD, respectively. These results show that the low-dimensional PSD is easier to reconstruct with high quality than STFT.

Table 9: **Comparison of position error and type prediction accuracy depending on the output representation.** Model-P denotes the model supervised by PSD reconstruction, while Model-S denotes the model supervised by STFT reconstruction.

| Model's output representation | Position Error (m)↓ | Type Acc.↑ |
|---|---|---|
| STFT | 2.03 | 0.57 |
| PSD | 1.44 | 0.73 |

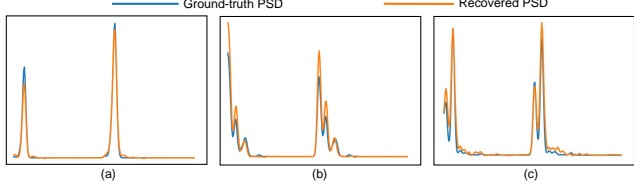

Figure 8: **Visualization of PSD Reconstruction.** We visualize the ground truth PSD in blue, while the network predicted PSD is shown in orange. In the three examples **(a)-(c)**, we find that the network can reconstruct the PSD with low error.

## A.6 Navigation with Ground-Truth Segmentation

In the experiments of the main paper, we find that some failures are caused by the inaccuracy of semantic segmentation, and thus cannot be attributed to the acoustic branch. In this experiment, we derive the ground-truth semantic segmentation from TDW, and use it in our modular planner in combination with our acoustic model. We test our method on a held-out set with 100 instances. Results in Table A.6 show that the success rate of our method can be further improved with the correct semantic mask.

Table 10: **Navigation Test with Ground-Truth Segmentation Mask.** Experiments show that the agent is more likely to successfully find the fallen object with ground-truth segmentation mask ("GT seg.").

| Method | SR ↑ | SPL ↑ | SNA↑ |
|---|---|---|---|
| Ours | 0.57 | 0.38 | 0.37 |
| Ours + GT seg. | 0.64 | 0.48 | 0.47 |

## A.7 Reproducibility

We will release the source code upon paper acceptance.

