# OpenReview forum: "Disentangled Acoustic Fields For Multimodal Physical Scene Understanding"
_ICLR.cc/2024/Conference — Submitted to ICLR 2024_

### Official Review · Reviewer_RUoW · 2023-10-24

**Soundness:** 2 fair
**Presentation:** 3 good
**Contribution:** 2 fair
**Rating:** 5
**Confidence:** 4

**Summary:**

This paper deals with the problem of predicting acoustic scenes such as types, materials, and positions of objects, and proposes a bi-directional encoder-decoder architecture for predicting acoustic scenes as well as synthesizing scene sounds from scene parameters. Since the proposed method assumes the existence of pairs of scene parameters and scene sounds as training examples, we can train both the encoder and the decoder in a supervised manner. This paper also presents a way of visualizing the prediction uncertainty of sound source locations and its application to agent navigation. Experimental evaluations with two public benchmarks demonstrate that the proposed method outperformed simple baselines.

**Strengths:**

1. The proposed architecture is interesting. If we can obtain lots of pairs of audio signals and object properties, the proposed architecture might be one of the best choices for simultaneously analyzing and synthesizing audio scenes.

2. Based on the experimental evaluations, the proposed method reasonably work well for several tasks.

**Weaknesses:**

1. The task of this paper is closely related to sound source localization. If the authors justify the effectiveness of the proposed method in terms of position errors of sound sources, experimental comparisons with several previous methods for this task would be mandatory. As presented in [Grumiaux+ JASA 2021 https://arxiv.org/abs/2109.03465], lots of previous methods have already been proposed for this purpose. I understand that many of those previous methods typically require multi-channel audio inputs and the proposed method employs a single-channel or binaural audio clip. However, you will find several techniques that work well only with single-channel audio, since single-channel sound source localization has been one of the typical tasks in audio signal processing before the deep learning era.

2. If we want to simply predict acoustic scenes from audio signals, we do not have to introduce decoders that generate audio signals from object properties. In this sense, the ablation study that compares the full proposed method and the one without decoders should be presented.

3. If the main objective of this paper is a proposal of a novel construction of neural acoustic fields, the proposed method should be compared with the original NAFs in the task of audio synthesis.

**Questions:**

Please check the above Weakness section.

---

> ### Author Response · Authors · 2023-11-17
> **Response to Reviewer RUoW**
>
> We are greatly encouraged by your assessment that our proposed architecture might be one of the best choices for simultaneously analyzing and synthesizing audio scenes. In line with your suggestions, we have conducted additional experiments to further demonstrate our method's superiority over baseline approaches. Additionally, we have refined our manuscript with more specific descriptions to clearly address each of your concerns. These updates, we believe, strongly validate the effectiveness of our approach. We address your concerns one by one as follows.
>
> > **Q1) Additional baselines in audio localization**
> >
> In response, we choose two strong methods after reading the survey on sound source localization methods [4] and implement them with pytorch. One feeds the STFT of the audio into U-Net[5], and the other feeds the original waveform into CNN [6]. To evaluate the performance of our model in sound source localization, we employed the REALIMPACT dataset[3]. The experimental results, presented in the table below, clearly demonstrate that our proposed model outperforms the established baseline methods. **We appreciate the guidance to substantiate our claims through comparative analysis and believe these additional results strengthen the effectiveness of our proposed approach.**
>
> | Method   | Angle Acc. | Height Acc. | Distance Acc.|
> |---------|---------|----------|-------|
> | Chance  | 0.100    | 0.067    | 0.250|
> | Unet + STFT [5] |  0.825    |  0.902  | 0.972  |
> | CNN + Waveform [6]  |  0.671    |   0.755   |  0.802|
> | Resnet + STFT [3] | 0.758    | 0.881    | 0.983|
> | Ours  | **0.900**    | **0.960**    | **0.994**|
>
> > **Q2) Ablation study with encoder only**
>
> We agree that the comparison of methods with and without the decoder component is crucial for evaluating the efficacy of the decoder in our model. But we would like to clarify that in Section 4.2 of our paper, **we have already conducted an analysis comparing the performance of our proposed method with a variant that does not include the decoder**. Experiments revealed that while the Success Rate (SR) remained comparable, there was a noticeable decrease in Shortest Path Length (SPL) and Sound Navigation Accuracy (SNA) metrics without the decoder.
>
> To address your concerns, we will enhance the clarity of this comparison in our manuscript, better articulating the critical role the decoder plays in the overall performance of our method.
>
> > **Q3) Comparison on audio synthesis**
>
>
> In response to reviewer RUoW's suggestion for a comparative evaluation with Neural Acoustic Fields (NAF) [1] in audio synthesis task, we collaborated with the NAF authors to create a baseline model equivalent to NAF. We then conducted a comparative analysis of our approach and NAF using 15,000 audio files from the REALIMPACT dataset[3]. This analysis focused on the accuracy of audio simulation, measured by comparing the Mean Squared Error (MSE) of the reconstructed Power Spectral Density (PSD) against the true PSD. The results, detailed in the accompanying table, clearly demonstrate our method's superior performance in audio simulation tasks compared to NAF, effectively addressing the reviewers' concerns.
>
>
> | Model   | NAF | Ours |
> |---------|---------|----------|
> | PSD MSE  | 0.0264    | 0.0052    |
>
>
>
>
>
> Thank you for the suggestions. We will include these details and change our notation to improve the clarity of our paper.
>
>
>
> [1] Luo, Andrew, et al. "Learning neural acoustic fields." NeurIPS (2022).
>
> [2] Gan, Chuang, et al. "Finding fallen objects via asynchronous audio-visual integration." CVPR(2022).
>
> [3] Clarke, Samuel, et al. "RealImpact: A Dataset of Impact Sound Fields for Real Objects." CVPR(2023).
>
> [4] Grumiaux, Pierre-Amaury, et al. "A survey of sound source localization with deep learning methods." The Journal of the Acoustical Society of America 152.1 (2022).
>
> [5] Patel, S., Maciej Zawodniok, and Jacob Benesty. "Dcase 2020 task 3: A single stage fully convolutional neural network for sound source localization and detection." DCASE2020 Challenge (2020).
>
> [6] He, Yuhang, Niki Trigoni, and Andrew Markham. "SoundDet: Polyphonic moving sound event detection and localization from raw waveform." ICML(2021).

---

> ### Author Response · Authors · 2023-11-21
> **Questions remaining?**
>
> Dear Reviewer RUoW,
>
> To streamline our responses:
>
> 1. **Q1) Additional Baselines in Audio Localization:**
>    - Implemented two strong baseline methods for sound source localization based on survey recommendations [4].
>    - Utilized the REALIMPACT dataset[3] for evaluation, showcasing our model's superiority over established baselines.
>    - Results demonstrate the robustness and effectiveness of our proposed approach in comparison to alternative methods.
>
>
> 2. **Q2) Ablation Study with Encoder Only:**
>    - Clarified that an ablation study comparing our method with and without the decoder component has already been conducted (Section 4.2).
>    - Acknowledged the critical role of the decoder in improving metrics such as Success Rate (SR), Shortest Path Length (SPL), and Sound Navigation Accuracy (SNA).
>    - Enhanced the manuscript to better articulate the significance of the decoder in our model's overall performance.
>
> 3. **Q3) Comparison on Audio Synthesis:**
>    - Collaborated with NAF authors for a comparative evaluation in the audio synthesis task, addressing the concern raised by reviewer RUoW.
>    - Conducted a thorough analysis using 15,000 audio files from the REALIMPACT dataset[3], demonstrating superior performance in audio simulation tasks compared to NAF.
>    - Results, as presented in the table, illustrate the improved accuracy of our method in audio synthesis.
>
> | Model   | NAF | Ours |
> |---------|---------|----------|
> | PSD MSE  | 0.0264    | 0.0052    |
>
> Thank you for your insightful feedback. If you have any remaining questions or concerns following our response, please let us know. We’d be very happy to do anything we can that would be helpful in the time remaining!

---

> ### Comment · Reviewer_RUoW · 2023-11-23
> **Thanks for the effort**
>
> Thanks for the effort in the rebuttal. I have read through all the author replies and I found that all the replies are reasonable. I think that this paper can be accepted if (1) all the author replies will be contained in the main paper (not in the supplementary material) and (2) there is a room for the acceptance.

---

### Official Review · Reviewer_reBa · 2023-10-28

**Soundness:** 2 fair
**Presentation:** 2 fair
**Contribution:** 2 fair
**Rating:** 5
**Confidence:** 4

**Summary:**

This work proposes a disentangled acoustic fields (DAF) as a complementary in acoustic modal for physical scene understanding. The DAF could potentially help embodied agent to construct a spatial uncertainty map over where the objects may have fallen.

**Strengths:**

1.	The main contribution of this paper is to enhance audio perception by so-called analysis-by-synthesis framework. The major strength is to maintain the (generated/synthesis one) power spectral density (PSD) consistency with the input audio.
2.	The downstream multi-modal planning experiments demonstrates the effectiveness of proposed framework.

**Weaknesses:**

1. The title “physical scene understanding” could be overclaimed the contribution since the audio perception is limited to constrained scenarios (e.g., fallen objects).
2. Though the DAFs or (the predict-generate) is novel in audio modality, it is not such innovative and is close to use the cycle-consistency to ensure robustness in vision modality. I would suggest this work more like an ICRA paper instead of ICLR.

**Questions:**

1. The title “physical scene understanding” could be overclaimed the contribution since the audio perception is limited to constrained scenarios (e.g., fallen objects).
2. Though the DAFs or (the predict-generate) is novel in audio modality, it is not such innovative and is close to use the cycle-consistency to ensure robustness in vision modality. I would suggest this work more like an ICRA paper instead of ICLR.

---

> ### Author Response · Authors · 2023-11-17
> **Response to reviewer reBa**
>
> Thank you for your constructive review. We carefully address your concerns as follow:
>
>
>
> > **Q1) A more informative title**
>
>
> We appreciate your suggestion regarding the title "physical scene understanding", given that our work primarily focuses on using audio information from objects impacting surfaces (which may not always be falling) to infer physical properties like location, size, and material. To more accurately represent the scope of our research, we propose modifying the title to "Disentangled Acoustic Field for **Physical Property Understanding"** This change better aligns the title with the goal and contributions of our work.
>
>
>
>
> > **Q2) Cycle-consistency in audio**
>
>
> Our utilization of the analysis-by-synthesis framework is fundamentally geared towards enhancing the model's comprehension of physical properties through audio perception, rather than focusing on robustness in vision. This approach is distinct from cycle-consistency in vision, as it specifically targets the underconstrained task of inferring and understanding acoustic scenes and their physical characteristics.
>
> > **Q3) Topic relevance.**
>
>
> We wish to emphasize that the core contribution of our work is the introduction of a Disentangled Acoustic Field (DAF), which significantly enhances the perception of an object's physical characteristics, including the material, type, and size.
>
> **Our research, while applicable to robotics, proposes an audio decomposition representation has broader applications in understanding the physical properties of objects that emit sounds**. It aligns with the broad themes in learned representations traditionally explored at ICLR. This assertion is substantiated by numerous precedents of similar research [3][4][5][6] being accepted and published at ICLR. Therefore, we firmly believe that our paper is aptly suited for ICLR, contributing to its diverse and innovative discourse.
>
>
>
>
> Thank you again for your advice and feedback. We will incorporate the suggestions into the paper.
>
> [1] Luo, Andrew, et al. "Learning neural acoustic fields." NeurIPS (2022).
>
> [2] Gan, Chuang, et al. "Finding fallen objects via asynchronous audio-visual integration." CVPR(2022).
>
> [3] Abdullah, Hadi, et al. "Demystifying limited adversarial transferability in automatic speech recognition systems." ICLR. 2021.
>
> [4] Ding, Shaojin, Tianlong Chen, and Zhangyang Wang. "Audio lottery: Speech recognition made ultra-lightweight, noise-robust, and transferable." ICLR(2021).
> [5] Shim, Kyuhong, Jungwook Choi, and Wonyong Sung. "Understanding the role of self attention for efficient speech recognition." ICLR(2021).
>
> [6] Lam, Max WY, et al. "BDDM: Bilateral denoising diffusion models for fast and high-quality speech synthesis." ICLR (2022).

---

> ### Author Response · Authors · 2023-11-21
> **Questions remaining?**
>
> Dear Reviewer reBa,
>
> Thank you for your constructive review. We appreciate your valuable feedback and have addressed your concerns as follows. To streamline our responses:
>
> 1. **Q1) A More Informative Title:**
>    - Proposed a modified title: "Disentangled Acoustic Field for Physical Property Understanding" to better reflect the focus on inferring physical properties from impacting sounds.
>
> 2. **Q2) Cycle-Consistency in Audio:**
>    - Clarified that our use of the analysis-by-synthesis framework aims to enhance audio perception for inferring physical properties, distinct from cycle-consistency in vision.
>
> 3. **Q3) Topic Relevance:**
>    - Emphasized the core contribution of our work, the Disentangled Acoustic Field (DAF), and its broader applications in understanding the physical properties of sound-emitting objects.
>    - Asserted the relevance of our research to ICLR, aligning with the diverse and innovative discourse present in previous ICLR publications [1][2][3][4][5][6].
>
> If you have any remaining questions or concerns following our response, please let us know. We’d be very happy to do anything we can that would be helpful in the time remaining!
>
> Thank you again for your time and thoughtful feedback.

---

> ### Comment · Reviewer_reBa · 2023-11-22
>
> Thanks for the authors’ efforts in rebuttal. Considering the contribution of this work and other reviewer's comments, I have decided to retain my score.

---

### Official Review · Reviewer_KbFD · 2023-11-04

**Soundness:** 3 good
**Presentation:** 3 good
**Contribution:** 3 good
**Rating:** 6
**Confidence:** 3

**Summary:**

The authors propose a novel approach to multimodal physical scene understanding that leverages a disentangled acoustic field model to capture the sound generation and propagation process. This approach enables the embodied agent to construct a spatial uncertainty map over where the objects may have fallen, which can be used to improve the success rate for the localization of fallen objects.

**Strengths:**

+ Transforming the sound from the waveform domain into power spectral density (PSD) representation rather than sound reconstruction is well-motivated.

+ The proposed disentangled acoustic fields (DAFs) are an interesting and technically sound model that explicitly disentangles sounds into several different acoustic factors.

+ DAFs can be used to infer the physical properties of a scene, represent uncertainty, and navigate and find fallen objects.

**Weaknesses:**

+ A video demo would be very helpful for us to understand the model's performance on the localization of fallen objects in the real world.


+ Currently, all of the experiments are conducted on synthetic datasets. It would be interesting to see how the model generalizes to real-world data.

+ The proposed method requires full labels to train DAFs. It would be beneficial to develop a self-supervised learning approach to avoid using many labels during model training.

+ Why not incorporate visual information into DAFs? Visual data can provide many acoustic cues.

**Questions:**

See the Weaknesses.

---

> ### Author Response · Authors · 2023-11-17
> **Response to Reviewer KbFD**
>
> We are strongly encouraged by your evaluation that our work is an interesting and well-motivated, and can technically perform well in multiple tasks. In response to the valuable feedback provided, we have taken several steps to address the concerns raised. First, we have produced demonstrative videos to visually showcase our model's effectiveness in localizing fallen objects, offering a clearer understanding of its real-world application. We have further revised the experimental sections of our manuscript to provide more clarity on our methods on real-world scenarios. We also acknowledge the importance of the suggestions regarding the potential of self-supervised learning to reduce dependency on labeled data, and the integration of visual information into our DAFs model. We discuss our approach to incorporating these elements in future work, aligning with the ongoing evolution of our research.
>
> We believe these revisions address your suggestions and open exciting avenues for the advancement of our work.
>
>
>
> > **Q1) Suggestion for a video demo**
>
>
> We strongly agree with the reviewer's suggestion for a video demonstration. We've created a video that showcases our model in action, with RGB images and semantic segmentation results during navigation. This website also includes a top-down view of the loss map and navigational trajectory, helping to illustrate how our model processes and interprets environmental data. We believe this video will effectively demonstrate the capabilities of our model in audio-visual navigation and object localization. For a detailed visual representation, we have provided the video link: [***https://sites.google.com/view/disentangled-acoustic-fields***](https://sites.google.com/view/disentangled-acoustic-fields).
>
>
>
> > **Q2) Experiments on real data**
>
> Thanks for your kind suggestion. We want to clarify that our study **has already used real-world data (REALIMPACT dataset [3]) for the object property task**. As detailed in Section 4.1 of our paper, the REALIMPACT dataset [3] contains 150,000 sound recordings from 50 different object categories. This provided a wide range of **real-world acoustic scenarios**. The results, shown in Table 2, demonstrate our method's improved accuracy in predicting physical properties like angle, height, and distance over baseline methods. This shows that our method generalizes well in real-world scenarios.
>
> To further clarify this point in our manuscript, we will augment this section with more explicit and detailed description, thereby addressing the concerns raised.
>
> > **Q3) Self-supervised learning**
>
> We deeply appreciate the reviewer's insightful suggestion regarding the implementation of a self-supervised learning approach for training our DAF. We agree that reducing the dependency on fully labeled data is a valuable and important direction for advancing the field.
>
> We would like to note that our system is supervised using the PSD, which is derived from the input. Our system is indeed self-supervised in this sense.
>
> However, we recognize the limitations this approach may have in terms of scalability in more diverse real-world scenarios. Moving forward, we are interested in exploring addition self-supervised learning paradigms. We believe that such an approach could significantly enhance the versatility and practicality of our method, enabling it to learn from a wider array of data with minimal labeling requirements.
>
> > **Q4) Visual information in DAFs**
>
> We agree with the reviewer's suggestion about the potential synergies between visual and audio information in enhancing environmental perception. This concept has been well-demonstrated in several studies [4]. For example, NAF [1] integrates sparse visual views with acoustic data to improve model performance.
>
> We would like to note our current setup uses visual information as a modular component of our DAF navigation task. We acknowledge that there remains additional opportunity in integrating visual information, and believe it is an important avenue for future research.
>
>
>
>
> [1] Luo, Andrew, et al. "Learning neural acoustic fields." NeurIPS (2022).
>
> [2] Gan, Chuang, et al. "Finding fallen objects via asynchronous audio-visual integration." CVPR(2022).
>
> [3] Clarke, Samuel, et al. "RealImpact: A Dataset of Impact Sound Fields for Real Objects." CVPR(2023).
>
> [4] Zhu, Hao, et al. "Deep audio-visual learning: A survey." International Journal of Automation and Computing 18 (2021).

---

> ### Author Response · Authors · 2023-11-21
> **Questions remaining?**
>
> Dear Reviewer KbFD,
>
> In light of your positive evaluation of our work as interesting, well-motivated, and technically proficient across multiple tasks, we've made concerted efforts to address your valuable feedback. Here's a succinct summary of our responses:
>
> 1. **Demonstration Videos and Experimental Clarity:**
>    - Produced videos to showcase our model's effectiveness in localizing fallen objects.
>    - Revised experimental sections for enhanced clarity on real-world scenarios.
>
> 2. **Specific Suggestions Addressed:**
>    - **Q1) Video Demo:**
>      - Created a [video](https://sites.google.com/view/disentangled-acoustic-fields) illustrating our model.
>
>    - **Q2) Experiments on Real Data:**
>      - Clarified the use of real-world data (REALIMPACT dataset [3]).
>      - Demonstrated improved accuracy in predicting physical properties.
>
>    - **Q3) Self-supervised Learning:**
>      - Acknowledged the importance and highlighted our system's inherent self-supervised nature.
>
>    - **Q4) Visual Information in DAFs:**
>      - Explained the integration of visual information and emphasized its importance in future research.
>
> These revisions aim to address your suggestions and pave the way for further advancements in our work.
>
> If you have any remaining questions or concerns following our response, please let us know. We’d be very happy to do anything we can that would be helpful in the time remaining!

---

### Official Review · Reviewer_oZCW · 2023-11-09

**Soundness:** 2 fair
**Presentation:** 2 fair
**Contribution:** 2 fair
**Rating:** 5
**Confidence:** 3

**Summary:**

The authors study the problem of multimodal physical scene understanding to infer the fallen objects' properties, direction and distance of an impact sound source. To deal with the limitation of current work NAFs which only captures the structure and material properties of a scene. They propose the disentangled acoustic fields to model acoustic properties across a multitude of different scenes. However, the design of DAF is incremntal to NAFs and the experiments are using only two different scenarios, kitchen and study room, which may not prove the generalization of DAF.

**Strengths:**

The problem is interesting.

**Weaknesses:**

1. The proposed work lacks novelty. The main contribution of this work is the introduction of DAFs, which are applied to the generative frameworks and multitask learning to address this task. Although it shows promising performance, it lacks some degree of innovation.
2. The paper mentions that NAFs lack generalization ability for new scenarios, while DAFs can effectively solve this problem. However, no relevant comparative experiments are observed in the experimental section.
3. The paper mentions that DAFs can address the generalization issue for different new scenarios, but the experimental section only demonstrates generalization for two different scenarios, kitchen and study room.

**Questions:**

The authors claim that their method is generalizable, why the experiments are using only two different scenarios, kitchen and study room?

---

> ### Author Response · Authors · 2023-11-17
> **Response to Reviwer oZCW**
>
> We thank Reviewer oZCW for the constructive review. We address specific comments below.
>
> > **Q1) Contribution and novelty**
>
>
> We would like to clarify that the relationship between our proposed DAF (Disentangled Acoustic Field) and NAF (Neural Acoustic Field) [1], as they tackle fundamentally different tasks. While it's true that DAF builds upon the foundational aspects of NAF, they are distinct in their application and capabilities. NAF primarily focuses on predicting audio propagation characteristics within a scene. In contrast, DAF extends this premise to a more complex task of inferring physical properties such as the location, material, and size of objects across unseen novel scenes, solely from audio information. This advancement represents a substantial shift in the application of acoustic modeling, enabling a broader scope of physical scene understanding beyond what NAF offers.
>
> Thus, while DAF and NAF share some conceptual lineage, their respective applications and the complexity of the tasks they address are fundamentally different, underscoring the innovative leap our work represents in the realm of audio-based property analysis.
>
> > **Q2) Comparative experiments against NAF.**
>
>
>
> We appreciate the reviewer's observation regarding the generalization ability tests. In response, we conduct a comparative evaluation with NAF [1] in audio synthesis task. we collaborated with the NAF authors to create a baseline model equivalent to NAF. We then made a comparative analysis of our approach and NAF using 15,000 audio files from the REALIMPACT dataset[3]. This analysis focused on the accuracy of audio simulation, measured by comparing the Mean Squared Error (MSE) of the reconstructed Power Spectral Density (PSD) against the true PSD. The results, detailed in the accompanying table, clearly demonstrate our method's superior performance in audio simulation tasks compared to NAF, effectively addressing the reviewers' concerns.
>
>
>
>
> | Model   | NAF | Ours |
> |---------|---------|----------|
> | PSD MSE  | 0.0264    | 0.0052    |
>
>
>
> > **Q3) Generalization across scenes**
>
> We would like to clarify that our method not only generalizes across two scenarios. As detailed in the experimental setup of Section 4.2 in our study, we employed Find Fallen Challenge [2], a comprehensive dataset encompassesing 30 distinct physical object types situated within **64 uniquely configured rooms**. Importantly, these rooms are categorized into two primary types: study rooms and kitchens. However, it is crucial to note that each category itself comprises 32 different room variations, offering a wide array of acoustic environments. Each scenario presents its unique set of acoustic challenges, thereby providing a robust testbed for evaluating the generalization capabilities of our DAF. The revised manuscript will include a more detailed explanation of these diverse experimental conditions.
>
> Moreover, our experiments follows [2] to utilize the same test split to assess the baseline algorithm's generalization performance. We believe that this extensive and varied dataset, encompassing a broad range of scenarios, adequately demonstrates the generalization prowess of DAF.
>
>
>
>
>
>
>
>
> We sincerely appreciate your comments. Please feel free to let us know if you have further questions.
>
> [1] Luo, Andrew, et al. "Learning neural acoustic fields." NeurIPS (2022).
>
> [2] Gan, Chuang, et al. "Finding fallen objects via asynchronous audio-visual integration." CVPR(2022).
>
> [3] Clarke, Samuel, et al. "RealImpact: A Dataset of Impact Sound Fields for Real Objects." CVPR(2023).

---

> ### Author Response · Authors · 2023-11-21
> **Questions remaining?**
>
> Dear R oZCW,
>
> To streamline our responses:
>
> **1. Differentiation from NAF:**
> - DAF and NAF serve different purposes; NAF focuses on audio propagation, while DAF extends to infer physical properties. This represents a significant advancement.
>
> **2. Comparative Experiments with NAF:**
> - Collaborated with NAF authors for a comparative audio synthesis task. Results show DAF's superior performance (MSE: 0.0052 vs. NAF's 0.0264).
>
> **3. Generalization Across Diverse Scenarios:**
> - DAF generalizes across 64 physically diverse rooms from the Find Fallen Challenge dataset.
>
> If you have any remaining questions or concerns, please feel free to let us know. We are eager to assist in any way possible within the given timeframe. Thank you for your time and consideration.

---

### Author Response · Authors · 2023-11-17
**General response (1/2)**

We sincerely appreciate the valuable feedback from all reviewers and are gratified by their positive evaluation of the effectiveness and significance of our work. Notably, **all four reviewers acknowledge the efficacy of our Disentangled Acoustic Fields (DAFs) in inferring multiple physical properties**, emphasizing its promising performance ('it shows promising performance' (oZCW)), technical soundness ('The proposed DAFs are an interesting and technically sound model that explicitly disentangles sounds into several different acoustic factors' (KbFD)), effectiveness ('experiments demonstrate the effectiveness of the proposed framework' (reBa)), and its potential as one of the best choices for such task ('might be one of the best choices for simultaneously analyzing and synthesizing audio scenes' (RUoW)).

We acknowledge the suggestions from all reviewers, and we are committed to addressing them comprehensively. Through clarifying our contribution and novelty, conducting additional experiments, and providing more detailed descriptions. To begin, we would like to offer a general clarification on the major concerns as follows:




## 1. General Clarification
### 1.1 Novelty and contribution of our work


First, in response to reviewer oZCW, we want to emphasize that our work on Disentangled Acoustic Fields (DAF) is fundamentally distinct from Neural Acoustic Fields (NAF)[1] in its core principles, motivations, and application scenarios. Unlike NAF [1], which focuses on audio reconstruction, our DAF approach specializes in inferring sound properties by explicitly decomposing and factorizing the latent space of the disentangled model. This further allows us to generate an uncertainty map, significantly enhancing our method's capability in navigation and exploration for the localization of fallen objects.

Second, to addressing reviewer reBa's comments about the scope of our submission, we clarify that DAF's applications extend beyond robotics. DAF is intrinsically aligned with broader questions in applying data-driven methods for audio analysis. This aligns well with the scope of ICLR, where numerous studies[2][3][4][5] in learned audio representation have been presented. Therefore, our work is well-suited for ICLR, as it significantly contributes to the continually evolving intersection of AI and audio processing. We appreciate the reviewer's attention to this aspect, and we want to assure them that the topic of our paper aligns closely with ICLR's focus. The decision to submit to ICLR was made after careful consideration and thorough analysis of the conference's scope and relevance to our research.



### 1.2 Generalization Capability



First, we would like to address reviewer oZCW's concern by emphasizing that our method demonstrates generalization across more than just two rooms. As detailed in the experimental setup of Section 4.2 in our study, we employed Find Fallen Challenge [6], a comprehensive dataset consisting of 8000 instances. This dataset encompasses 30 distinct physical object types situated within **64 uniquely configured rooms**. Importantly, these rooms are categorized into two primary types: study rooms and kitchens. However, it is crucial to note that each category itself comprises 32 different room variations, offering a wide array of acoustic environments. Each scenario presents its unique set of acoustic challenges, thereby providing a robust testbed for evaluating the generalization capabilities of our DAF.

Second, we sincerely appreciate reviewer RUoW's feedback regarding the need for comparative experiments with NAF[1]. In response, we reached out to the authors of NAF, implemented an equivalent baseline scenario, and conducted thorough tests on the REALIMPACT dataset[7]. The results will be elaborated in Section 2.2.





## 2. New experimental results

Reviewer RUoW suggests tests with more baseline methods on different tasks. To address his concern, we conducted two novel experiments to demonstrate that the superiority of our method.

---

> ### Author Response · Authors · 2023-11-17
> **General Response (2/2)**
>
> ### 2.1  Sound source localization task
>
> In response, we choose two strong methods for learned sound source localization [9] and implement them with pytorch. One feeds the STFT of the audio into U-Net[10], and the other feeds the original waveform into a CNN [11]. We train these models on a subset of the REALIMPACT dataset[3] an test them on a disjoint subset. The experimental results, presented in the table below, clearly demonstrate that our proposed model outperforms the established baseline methods. We appreciate the guidance to substantiate our claims through comparative analysis and believe these additional results strengthen the effectiveness of our proposed approach
>
>
>
> | Method   | Angle Acc. | Height Acc. | Distance Acc.|
> |---------|---------|----------|-------|
> | Chance  | 0.100    | 0.067    | 0.250|
> | Unet + STFT [10] |  0.825    |  0.902  | 0.972  |
> | CNN + Waveform [11]  |  0.671    |   0.755   |  0.802|
> | Resnet + STFT [3] | 0.758    | 0.881    | 0.983|
> | Ours  | **0.900**    | **0.960**    | **0.994**|
>
>
>
> ### 2.2  Audio synthesis task
>
> In response to reviewer RUoW's suggestion for a comparative evaluation with NAF [1] in audio synthesis task, we collaborated with the NAF authors to create a baseline model equivalent to NAF. We then conducted a comparative analysis of our approach and NAF using 15,000 audio files from the REALIMPACT dataset[7]. This analysis focused on the accuracy of audio simulation, measured by comparing the Mean Squared Error (MSE) of the reconstructed Power Spectral Density (PSD) against the true PSD. The results, detailed in the accompanying table, clearly demonstrate our method's superior performance in audio simulation tasks compared to NAF, effectively addressing the reviewers' concerns.
>
>
>
>
> | Model   | NAF | Ours |
> |---------|---------|----------|
> | PSD MSE  | 0.0264    | 0.0052    |
>
>
>
> ## 3. Demonstration Video
>
> We appreciate the reviewer's insightful suggestion regarding the inclusion of a demo video. In response, we have created a comprehensive video showcasing the agent's perspective using both the strong baseline approach (modular planning [6]) and our proposed DAF in two distinct environments, namely the study room and kitchen. The demonstration effectively illustrates the enhanced efficiency of our method in locating fallen objects, particularly with the aid of the uncertainty map. For a detailed visual representation, we have provided the video link: [***https://sites.google.com/view/disentangled-acoustic-fields***](https://sites.google.com/view/disentangled-acoustic-fields).
>
> ## 4. Additional clarifications and replies
>
> 1. Our study **has already used real-world data (REALIMPACT dataset [7])**. As detailed in Section 4.1 of our paper, the REALIMPACT dataset [7] contains 150,000 sound recordings from 50 different object categories. The results shown in Table 2 demonstrate our method's improved accuracy in predicting physical properties like angle, height, and distance over baseline methods. This shows that our method generalizes well in real-world scenarios.
> 2. Self-supervised learning for DAF is a promising direction. In our future work, we will explore self-supervised learning to make our approach more versatile.
> 3. We believe adding visual information to our Disentangled Acoustic Fields (DAF) could enhance DAF's capacity to understand complex environments, making it a good research topic for our future research.
> 4. The original title "physical scene understanding" might be misleading, we propose modifying the title to **"Disentangled Acoustic Field for Physical Property Understanding"**
> 5. In Section 4.2 of our paper, **we have already conducted an analysis comparing the performance of our proposed method with a baseline that does not include the decoder**.
> <!-- 6. In an upcoming version, we will revise our paper carefully, including incorporation of additional technical details elucidating our framework and an enhancement of the overall presentation.
>  -->
>
>
>
> ### Conclusion
> We thank the reviewers for their helpful feedback and suggestions for additional evaluation, which will make the paper substantially stronger. **We anticipate that these modifications will substantialy enhance the paper's quality and make it well-qualified for publication.**

---

> > ### Author Response · Authors · 2023-11-17
> > **Reference**
> >
> > [1] Luo, Andrew, et al. "Learning neural acoustic fields." NeurIPS (2022).
> >
> > [2] Abdullah, Hadi, et al. "Demystifying limited adversarial transferability in automatic speech recognition systems." ICLR. 2021.
> >
> > [3] Ding, Shaojin, Tianlong Chen, and Zhangyang Wang. "Audio lottery: Speech recognition made ultra-lightweight, noise-robust, and transferable." ICLR(2021).
> >
> > [4] Shim, Kyuhong, Jungwook Choi, and Wonyong Sung. "Understanding the role of self attention for efficient speech recognition." ICLR(2021).
> >
> > [5] Lam, Max WY, et al. "BDDM: Bilateral denoising diffusion models for fast and high-quality speech synthesis." ICLR (2022).
> >
> > [6] Gan, Chuang, et al. "Finding fallen objects via asynchronous audio-visual integration." CVPR(2022).
> >
> > [7] Clarke, Samuel, et al. "RealImpact: A Dataset of Impact Sound Fields for Real Objects." CVPR(2023).
> >
> > [8] Grumiaux, Pierre-Amaury, et al. "A survey of sound source localization with deep learning methods." The Journal of the Acoustical Society of America 152.1 (2022).
> >
> > [9] Grumiaux, Pierre-Amaury, et al. "A survey of sound source localization with deep learning methods." The Journal of the Acoustical Society of America 152.1 (2022).
> >
> > [10] Patel, S., Maciej Zawodniok, and Jacob Benesty. "Dcase 2020 task 3: A single stage fully convolutional neural network for sound source localization and detection." DCASE2020 Challenge (2020).
> >
> > [11] He, Yuhang, Niki Trigoni, and Andrew Markham. "SoundDet: Polyphonic moving sound event detection and localization from raw waveform." ICML(2021).

---

### Author Response · Authors · 2023-11-21
**Thank you and we are looking forward to your post-rebuttal feedback!**

Dear AC and all reviewers:

Thanks again for all the insightful comments and advice, which helped us improve the paper's quality and clarity.

The discussion phase has been on for several days and we have not heard any post-rebuttal responses yet.

We would love to convince you of the merits of the paper. Please do not hesitate to let us know if there are any additional experiments or clarification that we can offer to make the paper better. We appreciate your comments and advice.

Best,

Author

---

### Meta-Review · Area_Chair_kYm7 · 2023-12-11

**Metareview:**

**Summary**: This paper introduces "Disentangled Acoustic Fields" (DAF) for multimodal physical scene understanding, focusing on the localization of fallen objects through sound. DAF aims to infer object properties, direction, and distance from impact sounds, a notable shift from direct regression methods. The model theoretically captures sound generation and propagation, intending to create a spatial uncertainty map for object localization.

**Strengths**: The concept of using disentangled acoustic fields to enhance sound-based physical scene understanding is theoretically intriguing. The approach suggests a potential for more accurate object localization using sound and offers a fresh perspective in the realm of audio-visual navigation. The spatial uncertainty map concept presented in the paper is an interesting approach that could have practical applications.

**Weaknesses**: The paper's major drawback lies in its limited demonstration of the model's generalization capabilities, with experiments confined to just two types of scenarios. This narrow scope raised concerns about the model's applicability in diverse real-world conditions. Furthermore, the paper initially lacked comparative analyses with existing models like Neural Acoustic Fields (NAF), leaving its claimed superiority unverified. These shortcomings significantly impacted the paper's persuasiveness and overall acceptance.

**Justification For Why Not Higher Score:**

Despite the authors' efforts to address the concerns raised, the reviewers remained unconvinced about the paper's readiness for acceptance. The revisions and additional experiments were seen as insufficient to fully address the fundamental issues of generalization and comparative evaluation. Also, comparison to NAF was also missing in the initial version. As a result, the consensus leaned towards rejection (although it is still a borderline paper), with the feedback highlighting the need for more robust and extensive validation of the model's capabilities in diverse and realistic scenarios.

**Justification For Why Not Lower Score:**

NA

---

### Decision · Program_Chairs · 2024-01-16

Reject